# ARE EFFICIENT DEEP REPRESENTATIONS LEARNABLE?

**Maxwell Nye**
Massachusetts Institute of Technology
mnye@mit.edu

**Andrew Saxe**
Harvard University
asaxe@fas.harvard.edu

## ABSTRACT

Many theories of deep learning have shown that a deep network can require dramatically fewer resources to represent a given function compared to a shallow network. But a question remains: can these efficient representations be *learned* using current deep learning techniques? In this work, we test whether standard deep learning methods can in fact find the efficient representations posited by several theories of deep representation. Specifically, we train deep neural networks to learn two simple functions with known efficient solutions: the parity function and the fast Fourier transform. We find that using gradient-based optimization, a deep network does not learn the parity function, unless initialized very close to a hand-coded exact solution. We also find that a deep linear neural network does not learn the fast Fourier transform, even in the best-case scenario of infinite training data, unless the weights are initialized very close to the exact hand-coded solution. Our results suggest that not every element of the class of compositional functions can be learned efficiently by a deep network, and further restrictions are necessary to understand what functions are both efficiently representable and learnable.

## 1 INTRODUCTION

Deep learning has seen tremendous practical success. To explain this, recent theoretical work in deep learning has focused on the representational efficiency of deep neural networks. The work of Montufar et al. (2014); Pascanu et al. (2013); Telgarsky (2015); Mhaskar et al. (2016); Poole et al. (2016); Eldan & Shamir (2016), and Poggio et al. (2016), among others, has examined the efficiency advantage of deep networks vs. shallow networks in detail, and cite it as a key theoretical property underpinning why deep networks work so well. However, while these results prove that deep *representations* can be efficient, they make no claims about the learnability of these representations using standard training algorithms (Liao & Poggio, 2017). A key question, then, is whether the efficient deep representations these theories posit can in fact be learned.

Here we empirically test whether deep network architectures can learn efficient representations of two simple functions: the parity function, and the fast Fourier transform. These functions played an early and important role in motivating deep learning: in an influential paper, Bengio and LeCun noted that a deep logic circuit could represent the parity function with exponentially fewer terms than a shallow circuit (Bengio & LeCun, 2007). In the same paper, Bengio and LeCun pointed out that the fast Fourier transform, perhaps the most celebrated numerical algorithm, is a deep representation of the Fourier transform. Moreover, these functions have the sort of compositional, recursive structure which a variety of subsequent theories have shown are more efficiently represented in deep networks (Poggio et al., 2016). In particular, their deep representations require at least a polynomially smaller amount of resources than a shallow network would (from $O(2^n)$ neurons to $O(n)$ neurons for the $n$-bit parity function, and from $O(n^2)$ synapses to $O(n \log n)$ synapses for the $n$-point fast Fourier transform). Remarkably, despite the long history of the parity function as a motivating example for depth, whether deep networks in fact can learn it has never been directly investigated to our knowledge. We find that, despite the existence of efficient representations, these solutions are not found in practice by gradient descent. Our explorations reveal a puzzle for current theory: deep learning works well on real-world tasks, despite not being able to find some efficient compositional deep representations. What, then, is distinctive about real world tasks–as opposed to compositional functions like parity or the FFT–such that deep learning typically works well in practice?

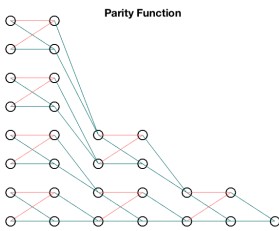 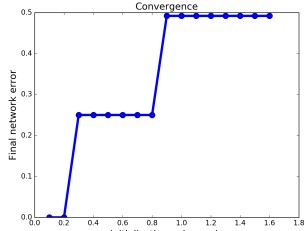 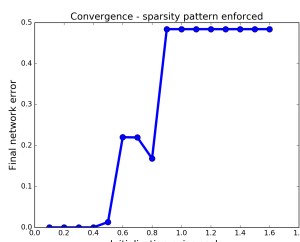

Figure 1: Learning the parity function. Left: Example hand-coded network for computing the parity function over $n = 8$ inputs using sigmoidal neurons (biases not shown). Each layer XOR's adjacent bits of the layer below. Note the recursive, compositional nature of the computation. Middle: Basin of attraction around hand-coded solution for $n = 32$. Deep sigmoid networks were initialized with the exact solution perturbed by Gaussian noise. The networks were then trained for 100000 minibatches of size 1000, each containing randomly drawn inputs. The resulting final test error was low only for small initialization noise, indicating that the efficient solution is a minimum, but is inaccessible from typical random initializations. Right: Basin of attraction with sparsity pattern enforced. Here the exact sparsity pattern for the parity solution was hard-coded into the weights, but the value of each nonzero weight was randomly drawn. The correct sparsity pattern improves the size of the basin of attraction, but training still fails from typical random initializations.

## 2   LEARNING THE PARITY FUNCTION

In our first experiment, we train a deep neural network to learn the parity function. The parity function on $n$ bits can computed with $O(n)$ computations. We hand-select weight matrices and biases, $\{W_i^{par}, b_i^{par}\}$, which exactly implement the parity function as a tree of XOR gates using sigmoidal neurons (see Fig. 1). We initialize a network with parameters $\{W_i^{par}, b_i^{par}\}$ and add scaled Gaussian noise (Glorot & Bengio, 2010). The variance of this noise controls the distance from a known optimal solution, and hence can be used to track the size of the basin of attraction around this optimum. Mean squared error is minimized using the Adam optimizer (Kingma & Ba, 2014) with learning rate $1e^{-4}$ and minibatches of 1000 randomly sampled binary vectors (other batch sizes yielded similar results). Our sigmoid activation function had inverse temperature parameter $\alpha = 10$. We tested networks with input size $n = \{16, 32, 64, 128\}$ and observed similar results for all sizes.

Figure 1 shows our results. We find that when the weights are initialized with only a small amount of noise, the network error converges to zero after training, and the network is able to learn the parity function. However, when initialized with higher noise, test error remains large and the network does not learn the parity function. We also perform a follow-up experiment, in which networks are trained and only the weights and biases which had nonzero values in the optimal solutions $\{W_i^{par}, b_i^{par}\}$ are allowed to vary with training. Under these conditions, the network has a much smaller search space and only needs to find the values of the sparse nonzero values. Again, we find that when the initialization noise levels are sufficiently high, the network does not learn the parity function. Hence, despite having a hyperefficient deep representation with compositional structure, a generic deep network does not typically find this solution for the parity function.

## 3   LEARNING THE FAST FOURIER TRANSFORM

In our second experiment, we train a deep linear neural network to learn the fast Fourier transform. Here we tried to create a best-case scenario for deep learning: whereas for the parity function we could only train on a subset of all possible inputs, here we exploit the linearity of the FFT to perform full gradient descent on the true loss function, corresponding to an infinite amount of training data. For this reason, failure to learn the FFT cannot be ascribed to insufficient training examples.

The fast Fourier transform (FFT) computes the discrete Fourier transform (DFT), a linear operation, with $O(n \log n)$ multiplications. We hand-select a set of weight matrices $\{W_i^{FFT}\}$ for a linear neural network which exactly implements the FFT for inputs of size $n$. We initialize our linear network at the hand-coded FFT solution and then add Gaussian noise. As above, the variance of this

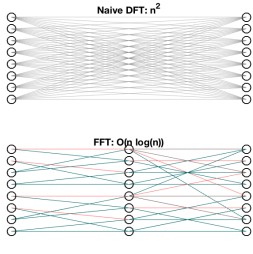 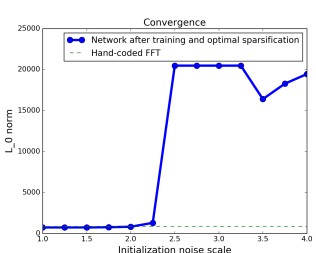 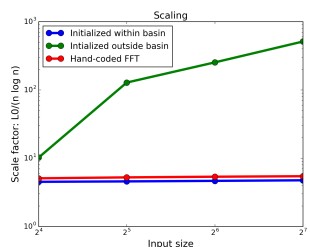

Figure 2: Learning the fast Fourier transform. Left, top: The discrete Fourier transform is a dense one layer linear network with $n^2$ nonzero synapses. Left, bottom: The fast Fourier transform is a deep linear network with just 2 nonzero connections per neuron. For larger inputs, the FFT can be applied recursively. Center: Basin of attraction around the hand-coded solution for $n = 32$. Only nearby initializations converge to the correct sparse solution. Right: Scaling behavior as a function of input size. The scaling factor is the network $L_0$ norm divided by $n \log n$, which is the complexity of the FFT. The correct asymptotic $O(n \log n)$ scaling corresponds to a flat line (as achieved by the hand-coded solution, red). Networks initialized near to the efficient solution (blue) successfully recover the correct scaling, but networks initialized farther away fail to scale as $O(n \log n)$ (green).

noise can be used to track the size of the basin of attraction around this optimum. The network is then trained to minimize the mean-squared error using the Adam optimizer. Without encouragement towards a sparse solution, a deep linear network will learn dense solutions in general. We encouraged sparsity with an $L_1$ regularization term, yielding the final loss: $Loss = \mathbb{E}\left[\|y - FFT[x]\|_2^2\right] + \beta \sum_j \|W_j\|_1^2$ where $\{x, y\}$ is a network input-output pair, $\beta$ is a regularization parameter, and $W_j$ is the $j$th weight matrix. We take the inputs to be drawn from a white distribution (e.g., $x \sim \mathcal{N}(0, I)$). We exploited the linear nature of the network model to perform our training. For linear networks, the gradient of the loss in expectation over all inputs can be calculated exactly by propagating a batch containing the $n$ basis vectors. We evaluated our neural network by examining both the final network error and the final sparsity pattern. In order to learn the fast Fourier transform, our neural network must have a low test error as well as a sufficiently sparse representation.

Figure 2 depicts our results. There exists a basin of attraction around the hand-coded optimal network configuration $W^{FFT}$, such that if the network is initialized close to $W^{FFT}$, setting small magnitude weights to zero actually decreases the test error, indicating convergence to the correct sparse solution. However, if the network is initialized outside this basin of attraction, any amount of weight sparsification causes the test error to increase, indicating that the solution is not sparse. Figure 2 right compares the $L_0$ norm of trained networks after optimal sparsification as a function of size for networks initialized near $W^{FFT}$ versus networks initialized far from $W^{FFT}$. Starting from random initial conditions, the learned networks do not achieve the $n \log n$ scaling of the FFT.

## 4    DISCUSSION

In this work, we studied functions which appear to differentiate efficient deep representation from deep learning. We emphasize that results on efficient deep representation are of critical importance and of independent interest. However, our results show that optimization remains a challenge in deep networks. The existence of a low approximation error deep representation does not mean that such a solution can be found, and theories based on the assumption of global empirical risk minimization may prove optimistic with respect to deep learning performance in practice. When treating these problems in the same way that practitioners do, we were unable to recover the efficient representations discussed in many theory papers. Our results point to a performance paradox: while in general efficient representations may be inaccessible by learning, deep networks work exceedingly well in practice. Possible resolutions are that (a) a subset of compositional functions are also easily learnable, (b) deep networks work well for another reason such as iterative nonlinear denoising (Kadmon & Sompolinsky, 2016), or (c) deep networks are fundamentally reliant on special architectures which impose the right compositional structure.

ACKNOWLEDGMENTS

A.M.S. thanks the Swartz Program in Theoretical Neuroscience at Harvard.

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
