# OpenReview forum: "Are Efficient Deep Representations Learnable?"
_ICLR.cc/2018/Workshop — Accept_

### Official Review · AnonReviewer3 · 2018-02-27
**Very good analysis**

**Rating:** 8
**Confidence:** 4

**Review:**

With simple examples, the authors clearly demonstrate the difference between what functions a deep neural network model can represent and what it can learn. This is important because the ability of deep neural networks to more efficiently represent complex functions is frequently cited as their primary advantage.

The presentation, experiments, and conclusions are very good.

---

> ### Author Response · Authors · 2018-04-18
> **Reply to Reviewer 3**
>
> We very much appreciate the positive feedback, thank you!

---

### Official Review · AnonReviewer2 · 2018-03-09
**testing theory of deep networks on learning simple functions**

**Rating:** 6
**Confidence:** 4

**Review:**

The paper does an empirical study of how well neural network optimized by back propagation learns simple functions: parity function/ fast fourier transform. The paper concludes that the deep network can not learn those simple functions unless initialized near the optimal solution.

The paper would benefit from explaining by what is meant by efficient representations (i.e number of neurons used vs number of examples etc).

Question regarding learning the  Fourier transform:
-  was the complex matrix learned by back propagation ? or you used the discrete cosine transform?
-  what if your signals in the training were translated version one of another? one would expect the network to learn the Fourier transform?
- In the Fourier case you use a deep linear network ,how many layers? what if one uses only a linear layer?
- was the network fully connected ? what if it was convolutional network? would it learn the fourier transform?

Studying those simple functions is insightful, but it is hard to draw conclusions out of this simple set of experiments. For instance why deep learning works in computer vision is due to the use of inductive biases such as CNN, for learning the Fourier transform one would expect that data augmentation with translation might help the learning.

---

> ### Author Response · Authors · 2018-04-18
> **Reply to Reviewer 2**
>
> Thank you for your helpful comments and feedback! We have compiled responses to your comments below.
>
> - In order to express the complex-valued fast Fourier transform with only real numbers, we simply concatenated the real and imaginary parts of each input and output vector. The hand-coded FFT weight matrices were correspondingly transformed. Thus the network weights and input were entirely real valued, but represent complex values.
>
> - The fast Fourier transform has a complexity of O(n log n), so any network with a single layer, while it may be able to implement the DFT in general, cannot implement the FFT.
>
> - The FFT networks were fully connected.
>
> - The reviewer notes that "why deep learning works in computer vision is due to the use of inductive biases such as CNN.” While we certainly agree that CNNs are extraordinarily useful for speeding up training, and are essentially necessary in practice, extensive experiments with MNIST seem to suggest that CNNs and fully connected nets don’t necessarily perform differently in the limit of large data and long training times (http://yann.lecun.com/exdb/mnist/, see “6-layer NN 784-2500-2000-1500-1000-500-10 (on GPU) [elastic distortions]” and “large/deep conv. net, 1-20-40-60-80-100-120-120-10 [elastic distortions]”). Because our data is synthesized and we trained many networks for millions of epochs, our experiments occur in this large data, long training regime. Thus, it is unclear to us whether inductive biases such as CNNs would have made a difference for our FFT experiments. Additionally, we showed that enforcing the correct sparsity pattern for the parity function does not appear to increase the ability of the network to converge.

---

### Official Review · AnonReviewer1 · 2018-03-12
**Interesting work benchmarking learnability of functions by deep networks against a ground truth**

**Rating:** 5
**Confidence:** 4

**Review:**

The premise of this paper is very interesting: the authors test the ability of deep networks to converge to the parity/FFT as a function of noise magnitude added to a ground truth initialization. They show that in both cases, the deep model only converges when initialized close enough.

My concern is with the lack of detail in some of the experimental settings: while I can completely believe that deep models do indeed behave in this way, I'd like to have an enumeration of the range of things tried before being convinced that this is the case.  I've mentioned some things that would be nice to see in the comments.

Major Comments
If the network architectures are as pictured in the figures, they look extremely small (in number of neurons), and it would be very interesting to have overparametrized networks as a comparison.

It seems important to also try other initialization schemes and see if models converge from there, particularly with overparametrized networks. My suspicion would be that for large enough networks, and small enough n, we would see the models correctly learn these functions.

More generally, it would be nice to see the range of architectures and methods tried described explicitly in the text (perhaps in the Appendix.)

Minor Comments:
Why use deep sigmoid networks for parity?
Test out on larger architectures?
Doesn't the DFT have very poor scaling (exponential variations in size?) This would be a reason why this might be hard for a deep network to learn?
Also, why skip to learning the FFT before seeing if the DFT can be learned?
Would like actual details of network architectures -- how big are they aside from the input size? (Width/depth?) Were different batch sizes tried? (1000 is a large-ish batch size.)
"Without encouragement towards a sparse solution, a deep linear network will learn dense solutions in general" -- even if many weights are nonzero, deep models typically have only a small set of weights that are very large in magnitude after training, so they are "almost" sparse?

I think this is very important and interesting work, but would like more details to paint a clearer picture before acceptance.

---

> ### Author Response · Authors · 2018-04-18
> **Reply to Reviewer 1**
>
> Thank you for your very thoughtful comments! We have compiled responses below.
> Many of the implementation details were present in earlier drafts of the paper, but were removed due to space constraints.
>
> - The Fourier transform networks had log(n/2) layers of size n by n, where n = {32, 64, 128, 256}. The parity function networks had input sizes n = {16, 32, 64, 128}, with 2 log(n) layers and the number of hidden units halved every other layer, such that the final output layer had a single neuron.
>
> - For the fast Fourier transform experiments, we tried initializing our networks with noisy weights around zero and the identity matrix, in addition to noisy versions of a hand-coded solution. We found that noisy versions of the hand-coded solution had the best convergence properties, while still not converging to a sparse solution if the initialization noise scale was too large. We thought this initialization to the hand-coded solution would represent a “best case” scenario, with other initializations likely to be less effective and this was empirically verified.
>
> - We used deep sigmoid networks for the parity function experiments simply because they are often used as the activation of choice when attempting to learn XOR gates. Since our hand-coded parity function baseline was made out of a tree of XOR gates, we thought sigmoid activations were the most natural choice. Future work could certainly conduct experiments with other activation functions.
>
> - For the fast Fourier transform experiments, we trained networks with hidden layers with up to 1.5x the number of units as the input and output layers. We observed no additional benefit, but it is possible that even greater overparametrization could help.
>
> - The DFT can be expressed as a single matrix multiplication, and can therefore be expressed by a linear neural network with no hidden layers. We chose to examine the FFT because it can be expressed by a deep network, but not by a shallow network, and our aim is to examine the learning properties of deep neural networks. Over the course of our experiments, we did indeed observe that a deep network with no sparsity constraint can learn to perform the DFT, but its weights will not generally have the efficient sparsity pattern of the FFT.
>
> - Sparsity was calculated in the following way: for each trained network, we found the maximum value \epsilon, such that if all weights with magnitude less than \epsilon were set to zero, the network error decreased. We defined the sparsity of the network as the L_0 norm of the network after rectification by this maximizing value of \epsilon. As reported in the paper, we found that there was a clear basin of attraction around the hand-coded solution, within which the network converged to a sparsity pattern consistent with O(n log n) scaling, and outside of which the network did not converge to a sparse pattern.
>
> - For the parity function experiments, we tried batch sizes of 250, 500 and 1000. We saw no significant differences between different batch sizes.

---

### Decision · Program_Chairs · 2018-03-20
**ICLR 2018 Workshop Acceptance Decision**

**Decision:**

Accept

**Comment:**

Congratulations, your paper was accepted to the ICLR workshop.